# Intraoperative Autofluorescence Imaging for Parathyroid Gland Identification during Total Laryngectomy with Thyroidectomy [note 1]

**DOI:** 10.3390/cancers15030875

**Published:** 2023-01-31

**Authors:** Raïs Obongo Anga, Muriel Abbaci, Joanne Guerlain, Ingrid Breuskin, Odile Casiraghi, Alix Marhic, Nadia Benmoussa-Rebibo, Héloïse de Kermadec, Antoine Moya-Plana, Stéphane Temam, Philippe Gorphe, Dana M. Hartl

**Affiliations:** 1Department of Head and Neck Oncology, Gustave Roussy Cancer Campus, 94805 Villejuif, France; 2Department of Head and Neck Cancer and ENT Surgery, Henri Becquerel Cancer Center, 76038 Rouen, France; 3Plateforme d’Imagerie et de Cytométrie, UMS AMMICa, Gustave-Roussy Cancer Campus—Grand Paris, Université Paris-Saclay, 94805 Villejuif, France; Laboratoire d’Imagerie Biomédicale Multimodale Paris Saclay, Université Paris-Saclay, CEA, CNRS, Inserm, 91401 Orsay, France; 4Department of Pathology, Gustave Roussy Cancer Campus, 94805 Villejuif, France

**Keywords:** parathyroid glands, autofluorescence, parathyroid reimplantation, laryngectomy, thyroidectomy, squamous cell carcinoma

## Abstract

**Simple Summary:**

Hypoparathyroidism is a known complication of total laryngectomy, although parathyroid preservation and/or reimplantation is not routine. Autofluorescence is a new technique for identifying parathyroid glands intraoperatively. The aim of this study was to evaluate the feasibility of autofluorescence in this context. We retrospectively reviewed 18 patients. Twelve had concomitant total thyroidectomy and 6 thyroid lobectomy. A median of 2 parathyroid glands were identified per patient, with 92% of them being found using the autofluorescent camera before visualization by the surgeon. Due to the wide excision for cancer all of the parathyroids were reimplanted, none showing tumor cells on intraoperative frozen section analysis. Only one patient (8%) had permanent hypoparathyroidism after 6 months and no cancer recurrence was observed in the sites of parathyroid reimplantation. Autofluorescence was an aid in finding parathyroid glands in these wide tumor resections and reimplantation of the devascularized glands was safe.

**Abstract:**

Objective: Hypoparathyroidism is a known complication of total laryngectomy, although parathyroid preservation and/or reimplantation are not routine. Autofluorescence is a new technique for identifying parathyroid glands intraoperatively. The aim of this study was to evaluate the feasibility of autofluorescence in this context. Materials and Methods: A retrospective study of patients undergoing total laryngectomy/pharyngectomy with concomitant thyroidectomy using the Fluobeam^®^ (Fluoptics, Grenoble, France) and frozen section of a parathyroid fragment in case of reimplantation. The rates of identification using autofluorescence, reimplantation, and hypoparathyroidism were evaluated. Results: Eighteen patients (16 males, median age 67) underwent total laryngectomy/pharyngectomy with total thyroidectomy (n = 12) or hemithyroidectomy (n = 6). A median of 2 parathyroid glands were identified per patient. Ninety-two percent were identified by autofluorescence before visualisation. All parathyroids were reimplanted due to devascularization. Temporary hypoparathyroidism occurred in nine patients, and was permanent in one patient. After 34 months of median follow-up (range 1–49), no tumor recurrence was observed in the reimplantation sites. Conclusions: To our knowledge, this is the largest study to evaluate autofluorescence during total laryngectomy with thyroidectomy. No tumor recurrence occurred in the sites of parathyroid reimplantation.

## 1. Introduction

Hypoparathyroidism, temporary or permanent, is a well-known complication of total thyroidectomy, occurring in up to 49% and 19% of patients, respectively [1,2,3,4,5]. Adding a total or even a hemithyroidectomy to total laryngectomy for squamous cell carcinoma significantly increases the risk for both temporary and permanent hypoparathyroidism, with an 8-fold higher risk of long-term calcium supplementation with total thyroidectomy and a 5.3-fold higher risk with hemithyroidectomy as compared to total laryngectomy alone [6]. Permanent hypoparathyroidism after total thyroidectomy has been shown to be associated with an increased risk of cardiovascular and renal morbidities, a higher risk of malignancies, and an increased risk of mortality [1,7]. Even when effectively treated, permanent hypoparathyroidism is still associated with a decreased quality of life [8]. Parathyroid identification, preservation, and reimplantation have been a standard of care in thyroid surgery for decades but have not been widely reported in the treatment of laryngeal squamous cell carcinoma (SCC), even though in a recent systematic review, Edafe et al. found rates of post-laryngectomy hypoparathyroidism ranging from 5–57% for temporary hypoparathyroidism and 0–12.8% for permanent hypoparathyroidism [9]. Classically, parathyroid autotransplantation is not a standard procedure in the treatment of squamous cell carcinoma due to the risk of transplanting malignant cells, although several retrospective studies attest to the feasibility, safety, and efficacy of this technique in reducing post-operative hypoparathyroidism [10,11,12,13,14].

Parathyroid autofluorescence is a relatively new technique for identifying and preserving parathyroid glands intraoperatively [15,16,17,18,19,20,21]. It is based on the presence of an endogenous fluorophore, yet to be identified, within the parathyroid glands that fluoresces when the parathyroid gland is illuminated with laser light of a certain wavelength (785 nm). The fluorescent light given off by the parathyroid gland can be picked up by a near-infrared camera. This enables the surgeon to better see the parathyroid gland that “lights up” as compared to the surrounding thyroid tissue. This technology has been shown to improve the rate of temporary hypocalcemia in the setting of total thyroidectomy, as well as the rate of inadvertent parathyroid resection and the need for parathyroid reimplantation [16].

The aim of this study was to evaluate the feasibility of using autofluorescence to identify parathyroid glands in the context of total laryngectomy with hemi- or total thyroidectomy. The secondary aims were to evaluate the oncologic safety of parathyroid reimplantation in these patients and the rate of temporary and permanent hypoparathyroidism.

## 2. Materials and Methods

A retrospective review of prospectively collected data from patients undergoing total laryngectomy with or without pharyngectomy and/or esophagectomy and concomitant hemi- or total thyroidectomy and uni- or bilateral paratracheal neck dissection with the use of the Fluobeam^®^ parathyroid autofluorescence system (Fluoptics^®^, Grenoble, France) in a comprehensive cancer center was performed. Prospective collection was performed between February 2018 and January 2022. Approval for the study was obtained from the institutional head and neck clinical research and ethics committee.

Concomitant total or hemithroidectomy was proposed in a multidisciplinary tumor board setting, according to the extensions of the tumor and according to current recommendations [22]. Thyroidectomy was indicated for tumors with subglottic and/or extralaryngeal extension or extraphyarngeal extension of piriform sinus tumors.

### 2.1. Surgical Technique

Parathyroids were identified in situ or on the operating specimen after resection using the Fluobeam^®^ (Fluoptics, Grenoble, France) system. During the thyroidectomy and paratracheal neck dissection, the surgeon attempted to visualize the parathyroid glands in situ, first without the autofluorescence camera and then with the camera. If all four parathyroids could not be visualized in situ, the resected specimen was explored ex vivo without and then with the autofluorescence camera. Analysis of the specimen was performed immediatedy after resection by an aide at a side table.

Tissue suspected to be parathyroid, with a fluorescent image, was analyzed by sending a 1–2 mm fragment to pathology for frozen section analysis, to confirm the diagnosis of parathyroid tissue and to ensure that no squamous cell carcinoma was present in the sample. Confirmed non-cancerous parathyroid tissue was then fragmented and auto-transplanted into the forearm, the pectoralis muscle, or one of the sternomastoid muscles.

### 2.2. Outcomes Measured

We identified the number of parathyroids visualized by the surgeon in situ before using the camera, and then the number visualized by the fluorescence camera in situ or on the operating specimen. We looked at the results of frozen section analysis for any falsely positive fluorescent tissue (frozen section negative for parathyroid tissue) and for any parathyroid tissue invaded by the tumor. We registered the number of parathyroid glands reimplanted and the location of reimplantation.

Each surgeon also noted the approximate time spent in the OR using the fluorescence camera.

We analyzed the rate of temporary hypoparathyroidism, defined as the need to take daily calcium and/or vitamin D supplementation up to 6 months after surgery, and permanent hypoparathyroidism as defined by therapy persisting after 6 months (Bergenfelz surgery 2020).

The rate of complications or cancer recurrence at the site of parathyroid reimplantation was recorded, as were patient outcomes at the last follow-up (cancer remission, recurrence, or death).

## 3. Results

Eighteen patients (16 males, median age 67, range 42–77) met the inclusion criteria. All underwent total laryngectomy with or without pharyngectomy, and four patients had concomitant resection of the cervical esophagus (Table 1). Twelve patients underwent total thyroidectomy and six underwent hemithyroidectomy, all with paratracheal neck dissection (bilateral n = 11, unilateral n = 7). Surgery was either salvage surgery or surgery in an already-irradiated field in sic cases. One patient (patient 12 in the table) had already undergone a total thyroidectomy for the primary tumor, and a laryngectomy was performed for local recurrence. Sixteen patients had SCC; one patient had tracheal adenoid cystic carcinoma and macroscopic invasion of the thyroid gland; and one patient had a high-grade muco-epidermoid carcinoma of the thyroid gland invading the larynx, pharynx, and cervical esophagus. Table 1 shows the details for each patient.

Seven different surgeons performed the surgeries. Intraoperative use of the fluorescence camera in situ took an additional 5–10 min. The analysis of the surgical specimen was performed in a sterile manner immediately after resection by an aide at a side table, which meant that the surgery was not halted or delayed by using the technique to look for resected parathyroids on the specimen ex vivo.

A median of two parathyroid glands were identified per patient in 14 patients (total number visualized n = 25, range 1–4 per patient), but in 4 patients no parathyroids were found, so in total an average of 25/30 or 0.83 parathyroids were found per thyroid lobe resected. A total of two parathyroid glands were visualized by the surgeon without the aid of the autofluorescence (Figure 1) and confirmed by autofluorescence imaging, but the remaining 23 were identified by autofluorescence alone before visualization by the surgeon. Nine were found in-situ and 14 were found ex-vivo on the specimen (Figure 2 and Figure 3). In one case, the fluorescent tissue corresponded to fat on the frozen section, but all of the other parathyroids were confirmed on the frozen section. No invasion by cancer cells was observed in the parathyroids analyzed. Due to devascularization concomitant with the tumor resection, all of the parathyroid glands analyzed were reimplanted. In one patient, despite finding fluorescent tissue, no parathyroids were analyzed because the tissue was considered to be too close to the paratracheal lymph node (patient 9, Table 1). Table 1 shows the number of parathyroid glands found by the pathologist at final pathology, as found in the pathology report.

Temporary hypoparathyroidism occurred in nine patients, with 75% of the 12 undergoing concomitant total thyroidectomy. Only one patient out of 15 evaluable at 6 months developed permanent hypoparathyroidism, for a total rate of 1/15 or 6.7%, or 1/12 (8.3%) who had total thyroidectomy. Of the eight patients with only temporary hypoparathyroidism, six were without supplementation after 1 postoperative month (patients 2, 3, 4, 6, 16, and 18; Table 1), and two recovered within 3 months (patients 9 and 11; Table 1). None of the patients undergoing thyroid lobectomy had temporary or permanent hypoparathyroidism. After a median of 34 months of follow-up (range 1–49 months), no tumor recurrence was observed in the sites of parathyroid reimplantation.

The rate of thyroid gland invasion on pathological analysis of the specimen was 2/16 cases of squamous cell carcinomas (12.5%), in addition to the patient with macroscopic invasion of the thyroid gland by adenoid cystic carcinoma of the trachea and the patient with mucoepidermoid carcinoma of the thyroid gland.

## 4. Discussion

Invasion of the thyroid gland by squamous cell carcinoma occurs in relatively rare instances of extensive laryngeal or pharyngeal tumors. A systematic review and meta-analysis of thyroid gland invasion in total thyroidectomy evaluated a total of 2177 patients, finding an 8% rate of pathological thyroid gland invasion. Risk factors in this study were: subglottic tumors or subglottic extension; T4 cancers; and thyroid cartilage invasion [23]. Similarly, Kumar et al. evaluated 16 studies with a total of 1180 cases of total thyroidectomy associated with total laryngectomy, finding thyroid invasion in 10.7% of cases [24]. Subglottic tumors and subglottic extension were risk factors. In a previous study at our institution, out of 182 patients undergoing laryngectomy with total or hemithyroidectomy, 12.6% had thyroid gland invasion on pathologic analysis, which was significantly associated with subglottic tumor extension [15]. The present study found thyroid gland involvement on pathological analysis in 2/16 (12.5%) of the cases of squamous cell carcinoma. Concomitant thyroidectomy has been shown to improve outcomes in these selected extensive laryngeal and hypopharyngeal cancers [22,25,26,27].

This low rate of thyroid gland invasion implies that parathyroid glands are likewise most probably rarely involved by tumor extension, although frozen section analysis does seem warranted before reimplantation to confirm the diagnosis of parathyroid tissue and the absence of tumor invasion.

Parathyroid identification, preservation, and reimplantation have been the standard of care in thyroid surgery for decades but are not widely reported when thyroidectomy is performed in the treatment of laryngeal squamous cell carcinoma, even though in a recent systematic review of 23 observational studies, Edafe et al. found rates of post-laryngectomy hypoparathyroidism ranging from 5–57% for temporary hypoparathyroidism and 0–12.8% for permanent hypoparathyroidism [9]. Tumor stage, salvage surgery, concomitant pharyngectomy, and concomitant esophagectomy have been shown to be associated with a higher risk of postoperative hypoparathyroidism [9,28,29]. A total laryngectomy with total thyroidectomy carries an 8-fold higher risk for permanent hypoparathyroidism as compared to a total laryngectomy alone, and a total laryngectomy even just with hemithyroidectomy carries a 5.3-fold increase in risk [6].

Historically, parathyroid autotransplantation was not recommended in surgery for squamous cell carcinoma because of the fear of transplanting malignant tissue, although there are reports as early as the 1970s of its successful implementation, particularly for hypopharyngeal and esophageal resections [10,11,12,13,14]. Every et al. recently published a series of 30 patients undergoing total laryngectomy or laryngopharyngectomy with parathyroid preservation or autotransplantation [14]. There were no tumor recurrences at the reimplantation sites. Out of 30 patients, four (13%) had temporary hypoparathyroidism and three (10%)—all 3 treated with salvage surgery after radiation therapy—had permanent hypoparathyroidism. Surprisingly, cases of permanent hypoparathyroidism occurred in the absence of total thyroidectomy. In our study, 50% of our patients had temporary hypoparathyroidism and 6.7% experienced permanent hypoparathyroidism, all occurring after total thyroidectomy. These high rates may be accounted for by the high number of patients undergoing concomitant pharyngectomy and/or esophagectomy with total thyroidectomy and the fact that 6 patients (33%) underwent salvage surgery after previous radiation therapy in our series.

Large population studies have shown that there are major health issues associated with chronic hypoparathyroidism. In a study of 4899 patients treated with total thyroidectomy for benign disease, Almquist et al. found on multivariate analysis that permanent hypoparathyroidism was an independent risk factor for increased overall mortality (adjusted hazard ratio 2.09, 95% confidence interval 1.04–4.20) [2]. Permanent hypoparathyroidism is also associated with an increased risk of cardiovascular and renal morbidities and a higher risk of malignancies [2,4], so preserving parathyroid function should be a concern when performing thyroidectomy associated with total laryngectomy, particularly due to the fact that laryngectomized patients already often have associated cardiovascular comorbidities. In the study by Bergenfelz et al., of the 4828 patients analyzed, those with pre-existing cardiovascular conditions and permanent hypoparathyroidism had a higher risk of cardiovascular events during follow-up than patients without hypoparathyroidism (hazard ratio 1.88; 95% confidence interval 1.02–3.47) [4]. Health and quality of life issues surrounding chronic hypoparathyroidism may be masked by other comorbidities such as functional impairment, anxiety, and depression in laryngectomized patients, leading to an underestimation of the clinical impact of permanent hypoparathyroidism [30,31,32,33,34,35].

Despite improvements in surgical technique, identification of parathyroid glands remains challenging due to their small size, variable anatomic locations, and variable aspects. Furthermore, laryngectomy for squamous cell carcinoma with hemi- or total thyroidectomy, with or without pharyngectomy or esophagectomy, entails wide resection, often with proximal ligation of the superior thyroid artery and extensive central compartment neck dissection [14]. Positive visual identification of parathyroid glands in the context of salvage surgery after chemoradiation may be almost impossible due to fibrosis and edema. In the systematic review by Edafe et al., patients with concomitant esophagectomy and total thyroidectomy, reported in 4/23 studies, had rates of transient and permanent hypoparathyroidism ranging from 62.1–100% and 12.5–91.6%, respectively [9]. Pharyngectomy, salvage surgery, and bilateral selective lateral neck dissection were also risk factors for a higher rate of hypoparathyroidism. Saito et al. also found a higher rate of permanent hypoparathyroidism in patients with esophagectomy and in patients who had had previous radiation therapy [28]. Three of the four patients in our study with esophageal resection only had temporary hypoparathyroidism (the fourth patient died from metastatic urothelial cancer within 6 months of the laryngectomy).

The fluorescent properties of parathyroid glands were discovered in the early 2000s by the Mahadevan-Jansen team at Vanderbilt University [17,18,19]. Without the addition of any exogenous fluorophore, that is, without injection of any substance, parathyroids, when illuminated with laser light of a wavelength of 785 nm spontaneously give off light in the near infrared spectrum and in particular with a peak at 820–830 nm that can be visualized with a near-infrared camera [18]. Normal parathyroids autofluoresce with a light intensity that is over 1.29 times higher than that of the surrounding thyroid tissue, which enables the surgeon to positively identify parathyroid glands [15,19]. Very few normal parathyroid glands (<2%) have a low fluorescent signal [21], but abnormal parathyroid glands in the context of parathyroid hyperplasia or adenoma generally have a more heterogeneous and less intense near-infrared signal [36,37].

In a pivotal multicenter prospective randomized trial including 245 patients undergoing total thyroidectomy, Benmiloud et al. demonstrated a reduction in temporary hypoparathyroidism from 22% to 9% when using autofluorescence technology. Parathyroid autotransplantation and inadvertent resection were lower in the autofluorescence group: 4% versus 16% and 3% versus 14%, respectively [16]. Other non-randomized studies have also shown that using this technology significantly decreases the rates of temporary hypoparathyroidism, inadvertent resection, and the need for parathyroid autotransplantation in thyroid surgery [38,39,40,41]. Finally, a recent meta-analysis of seven studies including 1480 patients found that autofluorescence significantly decreased the risk of inadvertent parathyroid resection in thyroid surgery and improved early postoperative rates of hypocalcemia [20].

To our knowledge, this is the second and largest study to evaluate the feasibility of using parathyroid autofluorescence technology to identify parathyroid glands in total laryngectomy with thyroidectomy. Barbieri et al. recently reported their experience using parathyroid autofluorescence in seven patients treated with total laryngectomy, associated with a total thyroidectomy in three patients and a hemithyroidectomy in four patients [42]. They identified 18/20 parathyroid glands, making an effort to dissect and preserve the glands in situ, which we did not do systematically in our study. The rate of hypocalcemia at 1 week was 42.8%, and at two weeks, 14.2% These results were better than those calculated from a historic control group. The rate of permanent hypoparathyroidism was not reported. Only one of our patients had permanent hypoparathyroidism after a 6-month follow-up, although our cohort is too small to draw any definitive conclusions, and we did not perform a comparative study with a control group.

Due to the extensive nature of oncologic resection in our cases, preservation in situ of parathyroids with their vascular pedicle was not attempted, and all of the parathyroids identified were reimplanted. The site of reimplantation was decided by the senior surgeon, and due to the retrospective nature of this study, there was no predefined or preferential site for reimplantation. Six of the 10 patients with PTH reimplantation into the sterno-mastoid muscle had further adjuvant radiation therapy. Four of these patients had concomitant total thyroidectomies and unilateral or bilateral central neck dissections. They all had temporary hypoparathyroidism but no permanent hypoparathyroidism. These numbers are too small to draw any conclusions, but it may be postulated that post-operative radiation therapy to the reimplanted parathyroid glands was not deleterious, at least in these patients. The delay between surgery and radiation therapy is classically 4–6 weeks, which may be sufficient for the reimplanted parathyroids to revascularize and begin functioning, allowing them to resist damage from the radiation [43,44]. However, other factors such as remaining hidden parathyroids or a fifth parathyroid gland may also have contributed to recovery of parathyroid function. This technique did not take much extra time to perform during the operation (5–10 min), but a prospective study will be needed to precisely analyze the added time needed to use this technology. The specimen once removed was analyzed on a separate table with the autofluorescence camera, and the parathyroids were retrieved by a surgical aide for analysis while the procedure continued. Parathyroid autotransplantation requires very little extra dissection [44,45]. We found only one case of a false positive, with frozen section analysis confirming the presence of a lymph node (not metastatic) and not parathyroid tissue. False positives seem to be rare but have been reported in cases of lymph node metastases in thyroid cancer, and brown fat may also auto-fluoresce significantly [21]. Falsely negative parathyroids, that is, those that have a low fluorescent signal, are also rare [21].

In four patients, the parathyroid glands were not found despite use of the autofluoresence technology, and in the other patients, globally fewer than 2 parathyroids were identified per thyroid lobe resected. This may be due to several factors. Given the proximal ligation of the inferior and superior thyroid arteries during total laryngectomy with thyroidectomy, the use of the near-infrared camera may not be as straightforward as in thyroid surgery. For the parathyroid to be visualized with the near-infrared camera, the surface of the gland needs to be exposed to the light. During total laryngectomy, the resection margins are wide and the parathyroid glands may remain deep under connective tissue and surrounding fat, whereas in thyroid surgery the resection is performed at the level of the thyroid capsule, more easily exposing the parathyroid glands to the light. There is a learning curve with this technology, as well. In order to obtain a better image, the lights in the OR need to be dimmed and the camera held very still [15]. In our study, seven different surgeons were involved. All had previously used the near-infrared camera, but some colleagues had more experience than others, and some had more experience in thyroid and parathyroid surgery, as well. Because our study was retrospective, our pathologists did not look specifically or systematically for parathyroid glands in the surgical specimens of these large resections. They reported parathyroid glands when they happened to see them. This probably partly explains why we did not find all of the parathyroid glands in all of the patients, either during surgery or on pathology.

Unfortunately, our cohort is small and heterogeneous. The clinical impact of using parathyroid autofluorescence needs to be further studied in prospective trials, which should include a cost-utility analysis of adding this technology to current practice.

## 5. Conclusions

To our knowledge, this is the first study to evaluate parathyroid autofluorescence technology in total laryngectomy with thyroidectomy. Parathyroid autofluorescence imaging is simple and allows for the identification of resected parathyroids, particularly in the operating specimen, that can be reimplanted. Parathyroid reimplantation was oncologically safe in our study. Due to the retrospective nature of our study and the small, heterogeneous cohort, we could not draw any conclusions regarding the impact of this technique on postoperative outcomes. Further studies are needed to optimize the use of this technology in this indication and further evaluate the clinical value of parathyroid identification and reimplantation in these patients.

## Figures and Tables

**Figure 1 cancers-15-00875-f001:**
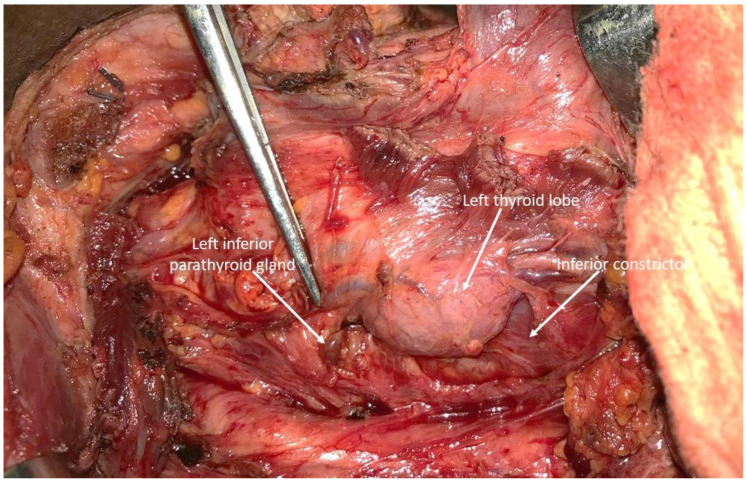
Intraoperative view of the left side of the neck showing the inferior parathyroid found in-situ by the surgeon on visual assessment.

**Figure 2 cancers-15-00875-f002:**
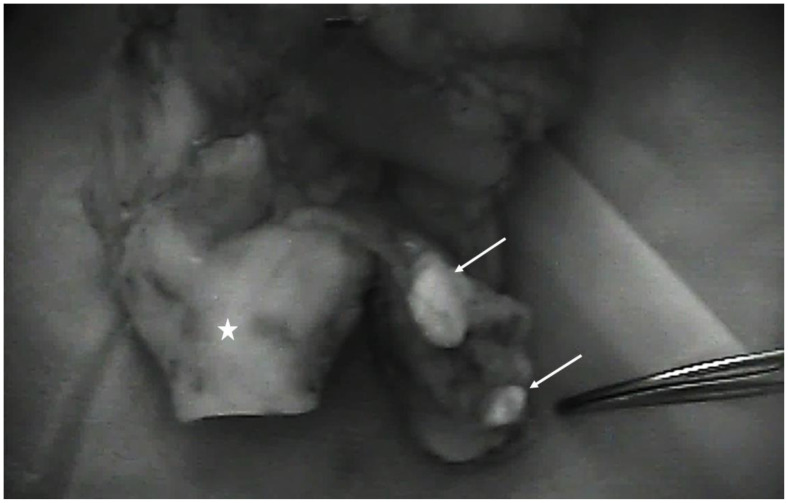
Posterior view of a total laryngectomy specimen (white star) seen with the near-infrared camera showing two autofluorescent parathyroids (white arrows).

**Figure 3 cancers-15-00875-f003:**
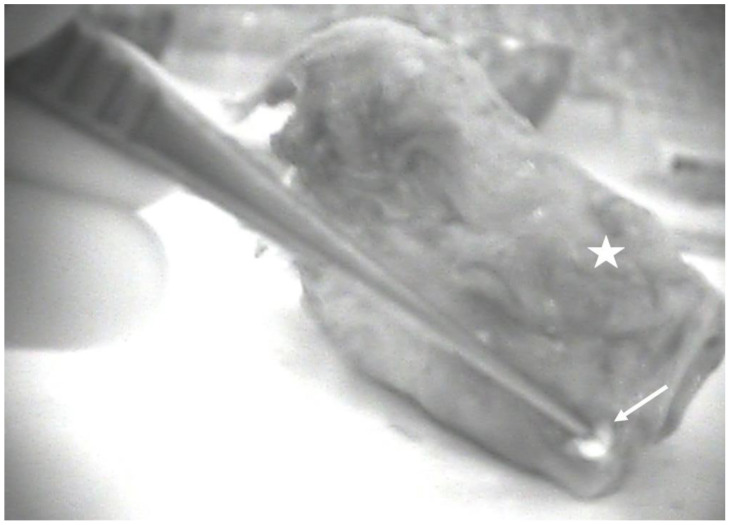
Lateral view of a total laryngectomy specimen (white star) seen with the near-infrared camera showing one autofluorescent parathyroid gland (white arrow).

**Table 1 cancers-15-00875-t001:** Demographics and results.

Patient	Sex Age	Pathology	Surgery	Thyroidectomy	Paratracheal Neck Dissection	PTH: Visual ID	PTH: Fluo ID	Total Number of PTH Reimplanted	PTH Found in Pathology Report	Temporary Hypo-PTH	Permanent Hypo-PTH	Postop RT	Follow-Up (Months)	Status
1	M 63	SCC	Larynx	total	bilateral	1 in situ	1 on specimen	2 (SCM)	1	No	NA	No	2.5	NED
2	M 70	SCC	Larynx, pharynx, esophagus	total	bilateral	0	2 on specimen	2 (SCM)	0	Yes	No	Yes	32	Distant metastases
3	M 67	SCC	Larynx	total	bilateral	0	2 on specimen	2 (pectoralis)	0	Yes	No	No	34	Local recurrence
4	F 46	Adenoid cystic carcinoma	Larynx, pharynx, esophagus	total	bilateral	0	1 on specimen	1 (forearm)	1	Yes	No	Yes	36	Lung metastases
5	M 53	SCC	Larynx, pharynx	hemi	unilateral	0	0	0	0	No	No	Yes	32	Local recurrence
6	M 76	SCC	Larynx	total	bilateral	0	4 in situ	4 (SCM)	0	Yes	NA	No	3	Death from distant metasastases
7	F 71	SCC	Larynx	hemi	unilateral	1 in situ	1 in situ	2 (SCM)	0	No	No	No	14	NED
8	M 77	SCC	Larynx	total	bilateral	0	1 on specimen	1 (pectoralis)	0	Yes	Yes	Yes	34	NED
9	M 75	SCC	Larynx, pharynx, esophagus	total	bilateral	0	0	0	3	Yes	NA	Yes	5	Death from urothelial carcinoma
10	M 67	SCC	Larynx	total	bilateral	0	0	0	1	No	No	Yes	36	NED
11	M 65	SCC	Larynx	total	bilateral	0	1 on specimen	1 (SCM)	0	Yes	No	Yes	45	NED
12	M 42	Muco-epidermoid carcinoma	Larynx, pharynx, esophagus	Total (in a previous operation)	bilateral	0	1 false positive	0	0	No	No	No	44	NED
13	M 66	SCC	Larynx, pharynx	hemi	unilateral	0	1 on specimen	1 (SCM)	0	No	No	Yes	4	Death from distant metastases
14	M 64	SCC	Larynx	hemi	unilateral	0	1 on specimen	1 (SCM)	1	No	No	No	47	NED
15	M 71	SCC	Larynx, Pharynx	hemi	unilateral	0	1 on specimen	1 (SCM)	0	No	No	Yes	49	NED
16	M 68	SCC	Larynx	total	unilateral	0	2 in situ 1 on specimen	3 (SCM)	0	Yes	No	Yes	5	NED
17	M 61	SCC	Larynx, pharynx	hemi	unilateral	0	2 in situ	2 (SCM)	0	No	No	No	1	NED
18	M 73	SCC	Larynx	total	bilateral	0	2 on specimen	2 (SCM)	1	Yes	No	Yes	8	NED

PTH: Visual ID: parathyroid identified by the surgeon and confirmed with autofluorescence; PTH: Fluo ID: parathyroid identified directly using autofluorescence; R: radiation therapy; SCC: squamous cell carcinoma; NA: not available; NED: no evidence of disease.

## Data Availability

Original data can be obtained by contacting the corresponding author.

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
