# Peer review of "Intraoperative Autofluorescence Imaging for Parathyroid Gland Identification during Total Laryngectomy with Thyroidectomyâ€"

_cancers, 2023, doi:10.3390/cancers15030875_

Round 1
Reviewer 1 Report
The authors performed a clinical study investigating the benefit of autofluorescence imaging during total laryngectomy with thyroidectomy to identify parathyroid glands and preserve parathyroid function.
The manuscript is well written, the references are timely and relevant. Some revisions are needed:
- A recent publication by Barbieri et al. (The use of near-infrared autofluorescence during total laryngectomy with hemi- or total thyroidectomy, PMID: 35920893, DOI: 10.1007/s00405-022-07584-4) is not included. In the manuscript is stated several times "up to our knowledge, this is the first study to evaluate autofluorescence during total laryngectomy with thyroidectomy". Please update.
- Table 1 reports 2 patients with permanent hypoparathyroidism (2/18 = 11.1%), whereas the manuscript text refers to 1 patient (p.6, l. 154; p.7, l. 203, and 1/18 would be =5.5%; p.8., l.262).
- Were any parathyroid glands left on the specimen without ATX (i.e. not visualised in the OR and detected in the pathology lab)? The manuscript text could state clearly if none was missed by use of the NIRAF exam, or if some were left undetected or on purpose on the specimen.
Minor comments:
- Move ethical approval statement to the first paragraph of Materials and Methods. Is there an ethical approval number to be included?
- The fluorescent light is picked up by a near-infrared camera (p. 2, line 57; p.8, l. 243)
- Format and number the Reference in p3, line 102 (Bergenfelz surgery 2020)
- ... abnormal parathyroid glands (p. 8, l. 246)
- Was the duration of the specimen examination with NIRAF by the aide recorded?
- Are the glands that were autotransplanted in the neck always included in the radiation field? How was decided between neck ATX and forearm ATX in this special context of laryngeal cancer and neck radiation?
Author Response
Dear Editor,
The authors thank the reveiwers for their time and effort and for their pertinent remarks that will improve our manuscript. We have made the modifications according to the reviewers’ remarks and have highlighted them in yellow in the revised text that we are submitting. Below, we respond to each comment.
Sincerely,
Dana Hartl
REVIEWER 1
The authors performed a clinical study investigating the benefit of autofluorescence imaging during total laryngectomy with thyroidectomy to identify parathyroid glands and preserve parathyroid function.
The manuscript is well written, the references are timely and relevant. Some revisions are needed:
A recent publication by Barbieri et al. (The use of near-infrared autofluorescence during total laryngectomy with hemi- or total thyroidectomy, PMID: 35920893, DOI: 10.1007/s00405-022-07584-4) is not included. In the manuscript is stated several times "up to our knowledge, this is the first study to evaluate autofluorescence during total laryngectomy with thyroidectomy". Please update.
- We have changed the abstract (conclusion) and added a description of this recent study in the discussion (pages 9-10, reference 42).
Table 1 reports 2 patients with permanent hypoparathyroidism (2/18 = 11.1%), whereas the manuscript text refers to 1 patient (p.6, l. 154; p.7, l. 203, and 1/18 would be =5.5%; p.8., l.262).
- We have modified the table and the text (patient 6 died 3 months postoperatively so there is missing data at 6 months). Three patients were not evaluable at 6 months for permanent hypoparathyroidism, so the true rate is 1/15 or 6.7% (page 6).
Were any parathyroid glands left on the specimen without ATX (i.e. not visualised in the OR and detected in the pathology lab)? The manuscript text could state clearly if none was missed by use of the NIRAF exam, or if some were left undetected or on purpose on the specimen.
- We have added a column in table 1 showing how many parathyroids were found by the pathologist on final pathology. Unfortunately, this was a retrospective study and the pathologists did not look specifically for parathyroid glands in these large resections. We have added a remark in the discussion (pages 6 and 11.)
Minor comments:
- Move ethical approval statement to the first paragraph of Materials and Methods. Is there an ethical approval number to be included?
- We have modified the text (page 4) and will provide the approval number to the journal, as they have requested.
- The fluorescent light is picked up by a near-infrared camera (p. 2, line 57; p.8, l. 243)
- corrected
- Format and number the Reference in p3, line 102 (Bergenfelz surgery 2020)
- corrected
- ... abnormal parathyroid glands (p. 8, l. 246) –
- corrected
- Was the duration of the specimen examination with NIRAF by the aide recorded?
- We did not record exactly how much time was needed during surgery or for the aide to find the parathyroids and we have added a comment in the discussion (page 10).
- Are the glands that were autotransplanted in the neck always included in the radiation field? How was decided between neck ATX and forearm ATX in this special context of laryngeal cancer and neck radiation?
- This was a retrospective study, so the site of reimplantation was decided by the senior surgeon. At the time, we did not have a standardized procedure (remark on page 10). Today we make an effort to reimplant outside of the potiential field of irradiation.

Reviewer 2 Report
Dear Editor,
I read with pleasure the article by Rais Obongo et al on « Intraoperative Autofluorescence Imaging for Parathyroid Gland Identification During Total Laryngectomy with Thyroidectomy » The authors report their experience of using autofluorescence to detect (and reimplant) parathyroid glands during laryngectomy with hemi or total thyroidectomy.
It is a well performed retrospective study on 18 patients. None of the 6 patients with hemi-thyroidectomy developped transient or permanent hypoparathyroidism (as expected), while 9 / 12 patients with total thyroidectomy developped transient hypopara and 2 / 12 definitive hypopara.
In the text (line 155), the authors describe one patient with definitive hypopara ; in the Table, patient 6 and patient 8 are described as hypoparathyroid. Please clarify whether one or two patient had definitive hypopara.
If laboratory values are available, it would be interesting for the readers to have a little more information on the recovery of parathyroid function in patients who had transient and not definitive hypopara (PTH levels POD 1 etc… until Month 6). If not available, perhaps the authors can tell approximately how long after surgery patients recovered.
The discussion on the recovery of parathyroid function could be a little bit expanded ; do the authors think that all patients who recovered did so thank to the reimplantation ?The only patient in whom 4 parathyroid glands were reimplanted apparently did not recover (patient 8)
Author Response
Dear Editor,
The authors thank the reveiwers for their time and effort and for their pertinent remarks that will improve our manuscript. We have made the modifications according to the reviewers’ remarks and have highlighted them in yellow in the revised text that we are submitting. Below, we respond to each comment.
Sincerely,
Dana Hartl
REVIEWER 2
Dear Editor,
I read with pleasure the article by Rais Obongo et al on « Intraoperative Autofluorescence Imaging for Parathyroid Gland Identification During Total Laryngectomy with Thyroidectomy » The authors report their experience of using autofluorescence to detect (and reimplant) parathyroid glands during laryngectomy with hemi or total thyroidectomy.
It is a well performed retrospective study on 18 patients. None of the 6 patients with hemi-thyroidectomy developped transient or permanent hypoparathyroidism (as expected), while 9 / 12 patients with total thyroidectomy developped transient hypopara and 2 / 12 definitive hypopara.
In the text (line 155), the authors describe one patient with definitive hypopara ; in the Table, patient 6 and patient 8 are described as hypoparathyroid. Please clarify whether one or two patient had definitive hypopara.
- We have modified the table and the text (patient 6 died 3 months postoperatively so there is missing data at 6 months). Three patients were not evaluable at 6 months for permanent hypoparathyroidism, so the true rate is 1/15 or 6.7% (page 6).
If laboratory values are available, it would be interesting for the readers to have a little more information on the recovery of parathyroid function in patients who had transient and not definitive hypopara (PTH levels POD 1 etc… until Month 6). If not available, perhaps the authors can tell approximately how long after surgery patients recovered.
- Unfortunately, due to the retrospective nature of the study, our biochemical data was incomplete. In addition, patients with hypoparathyroidism were supplemented on day 1, so calcemia levels normalized quickly. We did not routinely check parathormone levels. We have added the time to recovery, that is, the time after which patients no longer needed calcium/vitamin D supplementation on page 6.
The discussion on the recovery of parathyroid function could be a little bit expanded ; do the authors think that all patients who recovered did so thank to the reimplantation ?The only patient in whom 4 parathyroid glands were reimplanted apparently did not recover (patient 8)
- Our group of patients is very heterogenous, so it is difficult to say if recovery was due to reimplantation, a remaining hidden parathyroid or a supernumerary parathyroid gland (reported in 10-15% of cases). This is also the case for total thyroidectomy, hypoparathyroidism can be unpredictable. Reimplantation does not have a 100% success rate, unfortunately. Actually, patient 6 (all 4 glands reimplanted) was without supplementation at 1 month postoperatively, but died at 3 months (page 6). We have added a remark on page 10.
